# Cerebral Oxygenation Reserve: The Relationship Between Physical Activity Level and the Cognitive Load During a Stroop Task in Healthy Young Males

**DOI:** 10.3390/ijerph17041406

**Published:** 2020-02-21

**Authors:** Roman Goenarjo, Laurent Bosquet, Nicolas Berryman, Valentine Metier, Anaick Perrochon, Sarah Anne Fraser, Olivier Dupuy

**Affiliations:** 1Laboratoire MOVE (EA 6314), Faculté des Sciences du Sport, Université de Poitiers, 86000 Poitiers, France; roman.goenarjo@gmail.com (R.G.); laurent.bosquet@univ-poitiers.fr (L.B.); valentine.metier@gmail.com (V.M.); 2Department of Medical Physiology, Faculty of Medicine, Universitas Indonesia, Jakarta 10430, Indonesia; 3Department of Sports Studies, Bishop’s University, Sherbrooke, QC J1M 1Z7, Canada; nicolas.berryman@ubishops.ca; 4Centre de Recherche de l’Institut Universitaire de Gériatrie de Montréal, Montréal, QC H3W 1W5, Canada; 5Département des Sciences de l’Activité Physique, Faculté des Sciences, Université du Québec à Montréal (UQAM), Montréal, QC H2L 2C4, Canada; 6Laboratoire HAVAE (EA 6310), Département STAPS, Université de Limoges, 87032 Limoges, France; anaick.perrochon@unilim.fr; 7Interdisciplinary School of Health Sciences, Faculty of Health Sciences, University of Ottawa, Ottawa, ON K1N 6N5, Canada; sarah.fraser@uottawa.ca

**Keywords:** executive function, Stroop task, physical activity level, cerebral oxygenation, prefrontal oxygenation

## Abstract

*Introduction*: Many studies have reported that regular physical activity is positively associated with cognitive performance and more selectively with executive functions. However, some studies reported that the association of physical activity on executive performance in younger adults was not as clearly established when compared to studies with older adults. Among the many physiological mechanisms that may influence cognitive functioning, prefrontal (PFC) oxygenation seems to play a major role. The aim of the current study was to assess whether executive function and prefrontal oxygenation are dependent on physical activity levels (active versus inactive) in healthy young males. *Methods:* Fifty-six healthy young males (22.1 ± 2.4 years) were classified as active (n = 26) or inactive (n = 30) according to the recommendations made by the World Health Organization (WHO) and using the Global Physical Activity Questionnaire (GPAQ). Bilateral PFC oxygenation was assessed using functional near-infrared spectroscopy (fNIRS) during a computerized Stroop task (which included naming, inhibition, and switching conditions). Accuracy (% of correct responses) and reaction times (ms) were used as behavioural indicators of cognitive performances. Changes in oxygenated (∆HbO_2_) and deoxygenated (∆HHb) hemoglobin were measured to capture neural changes. Several two-way repeated measures ANOVAs (Physical activity level x Stroop conditions) were performed to test the null hypothesis of an absence of interaction between physical activity level and executive performance in prefrontal oxygenation. *Results:* The analysis revealed an interaction between physical activity level and Stroop conditions on reaction time (*p* = 0.04; ES = 0.7) in which physical activity level had a moderate effect on reaction time in the switching condition (*p* = 0.02; ES = 0.8) but not in naming and inhibition conditions. At the neural level, a significant interaction between physical activity level and prefrontal oxygenation was found. Physical activity level had a large effect on ΔHbO_2_ in the switching condition in the right PFC (*p* = 0.04; ES = 0.8) and left PFC (*p* = 0.02; ES = 0.96), but not in other conditions. A large physical activity level effect was also found on ΔHHb in the inhibition condition in the right PFC (*p* < 0.01; ES = 0.9), but not in the left PFC or other conditions. *Conclusion*: The results of this cross-sectional study indicate that active young males performed better in executive tasks than their inactive counterparts and had a larger change in oxygenation in the PFC during these most complex conditions.

## 1. Introduction

Participation in regular physical activity (PA) has been shown to reduce the incidence of many chronic diseases and mental disorders [1], therefore contributing to decreasing all-cause and cardiovascular mortality [2]. Regular PA has also been shown to influence cognitive performance throughout the lifespan [3], contributing to improved academic achievement in children, adolescents or young adults and preventing cognitive decline and dementia in older adults [4,5,6]. This beneficial effect on cognition seems to be more specific to executive functions, a subset of high-order cognitive functions including planning, inhibition or switching, which are mainly supported by the prefrontal regions of the cortex [7,8]. However, some studies reported that the association of physical activity on executive performance in younger adults was not as clearly established (as some have demonstrated no relationship) when compared to studies with older adults [9].

Several underlying mechanisms have been put forward to explain this link between regular PA and executive performance [10,11]. Cerebral oxygenation is considered as one of the cornerstones of this relationship [12]. Its importance is determined in part by the interaction between cerebral vasculature and the regulation of cerebral blood flow, which defines the magnitude of the cerebrovascular reserve, and by the capacity of cerebral blood vessels to increase cerebral blood flow (CBF) as a response to metabolic demand in activated brain regions [13,14,15]. Cerebral oxygenation is also determined by the availability of oxygen in the environment, as suggested by a study showing cognitive performance impairments in hypoxia [16].

Cerebral oxygenation reserve refers to the capacity of the prefrontal cortex blood vessels to increase blood flow in response to increased metabolic demands during increased neuronal activity in the prefrontal cortex. Several stimuli, such as cognitive tasks, can activate neurons leading to electrical and chemical activities. The immediate energy needs of activated neurons are met by oxidation of lactate present in the extracellular space, resulting in a transient decrease in lactate levels and oxygen concentrations [17]. As oxygen concentrations decrease, the lack of energy for adenosine triphosphate (ATP) synthesis via oxidative phosphorylation causes an increase in the level of extracellular adenosine and lactate. These conditions result in larger arteriole dilation and greater blood flow at the activated brain regions, thus providing a transient increase in oxyhemoglobin (ΔHbO_2_) and a relative decrease in deoxyhemoglobin (ΔHHb) due to the increased HbO_2_ concentration [18]. Neuroimaging technologies such as functional near-infrared spectroscopy (fNIRS) provide a measure of concentration changes in HbO_2_ and HHb, respectively, and allow for the investigation of the specific role of cerebral oxygenation during cognitive processing [19].

Although the studies in the functional near-infrared spectroscopy (fNIRS) field are growing exponentially [20,21,22], most of the studies that address the relationship between PA and cerebral oxygenation have the goal of measuring these factors in older adults to prevent or reduce cognitive impairment [23,24]. Little is known about the relationship between the participation in regular PA and cerebral oxygenation, more specifically prefrontal cortex (PFC) during cognitive processing in young adults, probably because the cerebrovascular reserve is not considered a limiting factor in this period of life [25]. Nevertheless, it is well established that the cardiovascular fitness of children and adolescents, and consequently of young adults, has been steadily decreasing for several decades, probably due to a significant decrease in weekly physical activity [26,27,28]. The potential consequences to cerebrovascular reserve are obvious (e.g., stroke, transient ischemic attack, cognitive issues, etc.), and suggest that the research effort on the link between PA and executive performance should not be limited to the elderly, but be concerned with all ages of life. The possibility of specific age-related effects of PA on executive performance should be explored. In all cases, mechanisms affecting cognitive performance in young adults should be studied in order to make appropriate recommendations. Therefore, the purpose of this study was to assess whether executive function and prefrontal oxygenation were dependent on physical activity level (active versus inactive) in healthy young males. We hypothesized that (1) executive performance would be better in participants with a higher level of PA compared to lower and that (2) in comparison to participants with a lower PA level, changes in PFC cerebral oxygenation would be greater in those with higher PA level in executive function conditions.

## 2. Method

### 2.1. Participants

Considering that PFC functions can be significantly influenced by fluctuations in the concentration of estrogen in women, we decided to focus specifically on males [29,30,31]. Fifty-six healthy young males (aged: 22.1 ± 2.1, range 18–26 years.) participated in this study. All participants signed a written statement of informed consent. They were non-smokers, did not undergo major surgery in the six months prior to the experiment, did not report any neurological or psychiatric disorders, and were not taking medication known to affect cognition. Moreover, given the physical implications of the study, participants were also screened and excluded for cardiovascular disease and moderate-to-severe hypertension based on self-report. The protocol was reviewed and approved by a national ethics committee for non-interventional research (CERSTAPS # 2017-23-11-17) and was conducted in accordance with recognized ethical standards and national/international laws.

### 2.2. Study Design

Participants completed all the tests within the same session. The session included a measure of physical activity levels, a cardiorespiratory assessment, as well as a cognitive assessment. The study design is presented in Figure 1.

### 2.3. Physical Activity Assessment

Physical activity was assessed using the French version of the Global Physical Activity Questionnaire (GPAQ). The French version of the GPAQ gives acceptable reliability and validity for the measurement of PA and sedentary time in adults [32]. The GPAQ was developed by the WHO for PA surveillance. It collects information on PA participation in three settings—PA at work, in recreational activities, and during travel to and from places, as well as sedentary behavior. The questionnaire consists of 16 questions covering both vigorous and moderate-intensity PA.

Based on the recommendation levels of physical activity for health from WHO (2009), participants were categorized as active, meeting the physical activity recommendations (≥150 min/week) or inactive (<150 min/week) [33].

### 2.4. Cardiorespiratory Fitness Assessment

The maximal continuous graded exercise test was performed on a motorized treadmill (Valiant 2 sport, Lode B.V., Groningen, Netherlands). The velocity was set at 5.6 km/h throughout the test, with an initial inclination of 0% grade. The inclination was increased by 6% after the first minute, and then by 2% every minute until voluntary exhaustion. Verbal encouragement was given every minute throughout the test [34]. Oxygen uptake (${\dot{V}O}_{2}$, in mL·kg^−1^·min^−1^) was determined continuously on a 30-s basis using a portable cardiopulmonary exercise testing system (MetaMax Cortex 3B, CORTEX Biophysik GmbH, Germany). Gas analyzers were calibrated before each test using ambient air and a gas mixture of known concentration (15% O_2_ and 5% CO_2_). The turbine was calibrated before each test using a 3-l syringe at several flow rates. The highest ${\dot{V}O}_{2}$ over a 30-s period during the test was considered the participant’s peak oxygen uptake (${\dot{V}O}_{2}\mathrm{peak}$, in mL·kg^−1^·min^−1^). Perceived exertion was assessed at the end of the test with the 15-point Borg scale (in which higher scores represent higher perceived exertion) [35].

### 2.5. Cognitive Assessment

The Computerized Modified Stroop task used in this study is based on the Modified Stroop Color Test and included three experimental conditions: naming, inhibiting, and switching [36]. Each block lasted between 2–4 min and was interspersed with 60-s resting blocks. Overall, there were three experimental task blocks (1 naming, 1 inhibition, and 1 switching) and 2 resting blocks, for a total length between 8–14 min. In total, there were 60 Naming trials (Block 1), 60 Inhibition trials (Block 2), and 60 Switching trials (Block 3). All trials began with a fixation cross (or square for switching condition) for 1.5 s, and all visual stimuli appeared in the center of the computer screen for 2.5 s. Participants responded with two fingers (index and major finger) from each of their hands on an AZERTY keyboard. In the Naming block, participants were presented with a visual stimulus of the name of colors (RED/BLUE/GREEN/YELLOW) in French presented in the color that is congruent with the word (i.e., RED presented in red ink). Participants were asked to identify the color of the ink with a button press. In the Inhibition block, each stimulus consisted of a color-word (RED/BLUE/GREEN/YELLOW) printed in the incongruent ink color (e.g., the word RED was presented in blue ink). Participants were asked to identify the color of the ink (e.g., blue). In the Switching block, in 25% of the trials, a square replaced the fixation cross. When this occurred, participants were instructed to read the word instead of identifying the color of the ink (e.g., RED). As such, within the Switching block, there were both inhibition trials in which the participant had to inhibit their reading of the word and correctly identify the color of the ink, and there were switch trials in which the participant had to switch their response mode to read the word instead of identifying the color of the ink when a square appeared before the word presented. Visual feedback on performance was presented after each trial. A practice session was completed before the acquisition run to ensure the participants understood the task. The practice consisted of a shorter version of the task. Dependent variables were reaction times (ms) and the number of errors committed (%). This task and procedure have been used successfully in previous studies [37,38].

The Stroop task is frequently used to examine executive function, as certain components of this task require extensive executive control [37,39,40]. The relationship between physical activity and executive function has been supported in systematic reviews, including studies that have measured executive functions with the Stroop task [41,42]. Those studies indicate that Stroop task can be used to observe the relationship between PA and executive function. Again, as studies about the relationship between PA and executive function in young adults, moreover using Stroop task, are limited, findings in this topic will be compelling.

### 2.6. PFC Oxygenation

The concentration changes of HbO_2_ (ΔHbO_2_) and HHb (ΔHHb) were acquired with the PortaLite fNIRS system (Artinis Medical Systems, Elst, Netherlands). This system utilizes near-infrared light, which penetrates the skull and brain but is absorbed by hemoglobin (Hb) chromophores in capillary, arteriolar, and venular beds [43]. The light was transmitted with two wavelengths, 760 and 850 nm, and data were sampled with a frequency of 10 Hz. The PortaLite uses wireless technology (Bluetooth), which allows participants to walk and move without the restriction of wires. Two sensors were placed on the forehead of the participants, one on the right and one on the left side. Both devices were positioned at the height of 10% of the nasion-inion distance from nasion, and the middle of the device was placed at 5% of the head circumference to the left and right from midline to avoid measuring the midline sinus. Those positions correspond to the Fp1 and Fp2 according to the international EEG 10–20 system and target left and right Brodmann’s areas 9 and 10, which roughly represent the dorsolateral and anterior prefrontal cortex (PFC). The sensors were shielded from ambient light with a black cloth and fixed with an elastic strap. Oxysoft version 3.0 software (Artinis Medical Systems, Elst, Netherlands) was used for data collection. This protocol for optode positioning has been used successfully in recent studies [44,45].

Using the different Hb absorption spectra, concentration changes of HbO_2_ and HHb in the PFC were calculated from the changes in detected light intensity. The calculation was conducted with the modified Lambert-Beer law, assuming constant light scattering [46]. The PortaLite has three transmitters and one receiver, with transmitter-receiver distances of 30, 35, and 40 mm. The change in prefrontal oxygenation was calculated as the average of the three channels of NIRS on the same side of the prefrontal cortex. NIRS data analysis was performed on unfiltered data. Each block of the task was calculated by averaging blocks on each task condition. The artifacts in the signals were identified by visual inspection and replaced by interpolation of adjacent data. The differential pathlength factor (DPF), which accounts for the increased distance traveled by light due to scattering, was set based on the age of participants. The differential pathlength factor (DPF) was specified for DPF807, which is determined using the formula: DPF807 = 4.99 + (0.067 × Age^0.814^) [47]. Variables of interest were relative changes in concentration of ΔHbO_2_, ΔHHb, and ΔtHb compared to the baseline (1 min at rest before the computerized Stroop task) [37,39,48]. Relative changes in concentration were measured because continuous-wave technology does not allow for the quantification of absolute concentrations due to its inability to measure optical path lengths [43,49].

### 2.7. Statistical Analysis

Standard statistical methods were used for the calculation of means and standard deviations. Normal Gaussian distribution of the data was verified by the Shapiro–Wilks test and homoscedasticity by a modified Levene Test. The compound symmetry, or sphericity, was assessed with the Mauchly’s test. Several t-tests for independent samples were used to explain the characteristics of PA groups. Several factorial analyses of variance (PA by Stroop condition) with repeated measures on Stroop conditions were used to test the null hypothesis of an absence of difference between groups. In presence of an interaction between PA level and Stroop conditions, a Bonferroni post-hoc test was used to further assess the relationship between PA level (active/inactive) on Stroop performance (naming, inhibition, switching) with behavioural (reaction times and accuracy) and neural (∆HbO_2_, ∆HHb and ΔtHb) outcomes. The magnitude of the difference was assessed by the Cohen’s d (*d*). The magnitude of the difference was considered either small (0.2 < *d* < 0.5), moderate (0.5 < *d* < 0.8), or large (*d* > 0.8) [50]. The significance level was set at *p* < 0.05 for all analyses. All calculations were made with Statistica 7.0 (StatSoft, Tulsa, OK, USA).

## 3. Results

Participants’ characteristics are presented in Table 1. A large difference was found in GPAQ scores (*t*
_(54)_ = 6.35, *p* < 0.001, *d* = 1.77) and in ${\dot{V}O}_{2}\mathrm{peak}$ (*t* (54) = 5.31, *p* < 0.001). There were no other significant differences between groups.

### 3.1. Computerized Stroop Task Results

The 2×3 repeated-measures ANOVA on Stroop reaction times revealed a marginal effect of physical activity level (*F*
_(1, 54)_ = 3.72, *p* = 0.06), in which active individuals had faster overall reaction times than inactive individuals (726 ms ± 95 and 784 ms ± 82, respectively; *d* = 0.65). There was also a main effect of the Stroop condition (*F*
_(2, 54)_ = 225.66, *p* < 0.001). Participants completed the naming condition faster than inhibition and switching conditions (605 ms ± 84, 707 ms ± 92, and 958 ms ± 146, respectively, 1.17 < *d* < 2.99). These main effects are superseded by a significant physical activity level by Stroop condition interaction (*F*
_(1, 54)_ = 3.40, *p* = 0.04). Bonferroni corrected post-hoc analysis revealed a moderate difference in reaction time in which active individuals were faster than inactive individuals in the switching condition (901 ms ± 146 and 1008 ms ± 138, respectively; *p* = 0.05; *d* = 0.77). We found no difference in reaction time between active and inactive individuals for the naming and inhibition Stroop conditions.

The 2×3 repeated-measures ANOVA assessing differences in accuracy on the computerized Stroop task revealed a significant main effect of the Stroop condition (*F*
_(2, 54)_ = 40.65, *p* < 0.001). Participants completed naming and inhibition conditions with significantly higher accuracy than the switching condition (96.7% ± 2.2, 97.4% ± 2.4, and 92.6% ± 4.2, respectively, 0.29 < *d* < 1.41). There were no effects of physical activity level on accuracy and no significant interactions. These results are presented in Table 2.

### 3.2. PFC Oxygenation Results

Total PFC oxygenation changes during the Stroop task are presented in Figure 2, and the details for the right and left hemispheres are presented in Table 3.

For ΔHbO_2_, the 2×3 ANOVAs indicated a significant main effect of the Stroop condition in the left PFC (*F*
_(2, 54)_ = 44.78, *p* < 0.001) and the right PFC (*F*
_(2, 54)_ = 72.89, *p* < 0.001). This main effect was superseded by significant interactions of physical activity level by the Stroop condition in the left PFC (*F*
_(2, 54)_ = 6.77, *p* < 0.01) and right PFC (*F*
_(2, 54)_ = 16.05, *p* < 0.001). Post-hoc analysis revealed a large difference in which active individuals had greater ΔHbO_2_ than inactive individuals in the switching condition (6.16 μmol·L^−1^ ± 2.38 and 3.73 μmol·L^−1^ ± 2.67, respectively; *p* = 0.05; *d* = 0.77). When examining cognitive load between the naming and inhibition condition, there was a greater ΔHbO_2_ in the right and left PFC in both physical activity groups. However, between the inhibition and switching condition, we found greater ΔHbO_2_ in the right and left PFC in active individuals but not in inactive individuals.

For ΔHHb, the 2×3 ANOVA revealed a main effect of the Stroop condition in the left PFC (*F*
_(2, 54)_ = 4.07, *p* = 0.02) and in the right PFC (*F*
_(2, 54)_ = 4.57, *p* = 0.01). We also found an interaction of physical activity level by Stroop condition in the right PFC (*F*
_(2, 54)_ = 3.25, *p* = 0.04). Post-hoc analyses demonstrated that between the naming and inhibition condition, there was greater ΔHHb in right PFC in active individuals but not in inactive individuals.

The typical response of cerebral oxygenation during the Stroop procedure is presented in Figure 3.

## 4. Discussion

The aim of the current study was to assess whether executive function and prefrontal oxygenation were dependent on physical activity level (active versus inactive) in healthy young males. Based on the existing literature, we hypothesized that the physical activity level would selectively enhance performance in executive conditions (inhibition and switching conditions) in a computerized Stroop task. Secondly, we hypothesized that in comparison to participants with a lower PA level, changes in PFC cerebral oxygenation would be greater in those with higher PA level in executive function conditions. The results of this study supported our first hypothesis, as we found that active individuals performed better in the executive condition of the computerized Stroop task than inactive individuals. However, this effect was specific to the switching condition (the most complex condition) and did not emerge in the inhibition condition. Regarding our second hypothesis, we found a greater amplitude response in PFC oxygenation during the switching condition of the Stroop task in active young males when compared to inactive.

For the cognitive performance findings, both groups had higher accuracy scores and shorter reaction times in the naming condition compared to the inhibition and switching conditions. This finding indicates that the non-executive condition (naming condition) was less demanding than the executive conditions and the manipulation of the cognitive load was successful as the inhibition and switching conditions require executive functions and the naming condition does not. The Stroop condition effect in our study is in accordance with the Stroop task findings in healthy younger and older females [37]. When compared to the overall cognitive performance, the active group had the same accuracy but with shorter reaction times than the inactive group. More specifically, this difference in reaction time performance emerged in the switching condition but not in the naming and inhibition conditions. More importantly, we also found an interaction of physical activity level by Stroop condition in reaction times, such that the increasing executive function load resulted in longer reaction times for the inactive individuals compared to active individuals. These data suggest that when the condition requires greater executive control, the active individuals demonstrate better executive functioning in the form of faster performances than inactive individuals. These results are in line with a longitudinal study that reported a specific physical activity level effect on the switch and mixing costs in task switching in young adults [51]. Further, consistent with other findings using the Stroop task, this result is in accordance with a cross-sectional study of younger adults and older adults and a longitudinal study of older adults [52].

In line with our previous study that examined younger and older females, no interaction of physical activity level by condition in the accuracy performance was found in our participants. This suggests that the computerized Stroop task is more sensitive to reaction time changes than accuracy when testing the effect of physical activity level on Stroop performances. All these results suggest that fulfilling physical activity recommendations (of ≥ 150 min/week) has a positive impact on cognition in healthy young males, specifically in the executive function domain.

Moreover, we found no difference between the groups in inhibition performance. As the inhibition performance has a lower cognitive load than the switching condition [53], it is possible that the inhibition condition was not cognitively challenging enough for healthy young males. Our result is in line with another study in older adults by Coubard et al. that evaluated executive functions after contemporary dance training and demonstrated an executive function improvement after contemporary dance training that was seen only in a switching condition (rule shift cards sorting test) but not in an inhibition condition (Stroop task) [54]. The best of our knowledge, this is the first study that compared the effect of physical activity level on the inhibition condition and switching condition in healthy young males.

Regarding PFC oxygenation, both groups had greater ΔHbO_2_, ΔHHb, and ΔtHb, on the executive conditions when compared with the naming condition. This result supports our cognitive load manipulation since participants increased PFC oxygenation in the more cognitively demanding executive conditions. This type of increased cerebral oxygenation to increasing task demands has already been reported in the younger adult literature [55,56,57]. In addition, we also found an interaction between physical activity level and Stroop condition when evaluating PFC oxygenation changes. This interaction demonstrated that increased executive function load during inhibition and switching conditions resulted in greater ΔHbO_2_ in left and right PFC, greater ΔHHb in the right PFC, and greater ΔtHb in left and right PFC in active individuals when compared with inactive individuals. These results suggest that active individuals may have a higher capacity to regulate the PFC oxygenation necessary to deal with increasing cognitive demand in the computerized Stroop task, hence proposing a mechanism in which physical activity can positively affect executive performances. Beyond this potential link between PA level, PFC oxygenation, and cognitive performance, the scientific literature of the last decade has also suggested a similar link with cardiorespiratory fitness (CRF) [24,37], which is regularly considered as one of the determinants of PA level [58,59]. In our study, physically active participants also displayed greater CRF (as measured by peak oxygen uptake) and presented a higher PFC oxygenation during a cognitive task. These results are in accordance with previous studies which reported that participants with the highest CRF also presented a higher PFC or occipital cortex oxygenation during a cognitive or visual task, whatever the age, thus supporting the cerebrovascular reserve hypothesis [24,37,60]. Agbangla et al. found a similar pattern in working memory task (n-back) and reported that high-fit older adults had a higher response to PFC oxygenation than low-fit older adults [23]. More specifically, active individuals displayed a greater decrease of ΔHHb then inactive individuals during the inhibition condition, which represents greater oxygenation. A previous fNIRS study reported that the right inferior gyrus of the prefrontal cortex, which is known to contribute to the inhibition process, was more active during an executive task in higher-fit individuals [37]

Theoretically, the present results combine some hypotheses from the previous studies, such as the compensation-related utilization of neural circuits hypothesis (CRUNCH), which stipulates that some cerebral regions will be more activated as specific task load increases, such as astrocyte-neuron lactate shuttle (ANLS), and cerebrovascular reserve [14,18,61]. In the CRUNCH model, cerebral activation should increase until a certain cognitive capacity is reached [61]. Afterward, cerebral activity may plateau or decrease, either because there are no further resources available or because there is no performance benefit in using brain resources any further. Additionally, according to the ANLS mechanism, the dynamics of cerebral activity is coupled with energy metabolism in the brain [17]. As a consequence, cerebral activation could be restricted by the availability or reserve of metabolic substances in the brain (e.g., oxygen, glucose, and lactate). The availability of these substances in the brain depends on brain perfusion and how cerebral blood vessels respond to metabolic demand. The cerebrovascular reserve hypothesis proposes that cerebrovascular control is positively associated with cognitive function [14]. Thus, a higher physical activity level, which is related to better cerebrovascular health, should be associated with better cognitive function.

In this study, we found that the interaction in cognitive performance, more specifically reaction time performance, was concurrent with the pattern of PFC oxygenation between physical activity level groups. One important finding to note from the present study is that inactive individuals had a poorer performance in the switching condition, which is the most complex condition, than their active counterparts but not in the other conditions. In parallel, there was an absence of increased oxygenation in response to the increased complexity of the Stroop task during the switching condition in inactive individuals. It is possible that inactive individuals had reached their cognitive capacity limit in a less complex condition than active individuals, and this alteration may be due to a lower capacity to increase PFC oxygenation in this group. The difference in the PFC oxygenation could be explained by the difference in cerebrovascular control as postulated in the cerebrovascular reserve theory.

### Limitations

As reported in a recent study by Weis et al. (2019), brain connectivity differs across menstrual cycle in females but remains stable in males, especially when frontal areas are involved [62]. Moreover, information on young adult males is still lacking, while several studies have been published on females [24,37]. This explains why this study focused on males. However, this approach limits the generalizability of our findings to this specific population. Understanding sex specificities is of great importance. Although several studies have been conducted with males or females separately, differences in participant’s characteristics (other than sex), as well as in tests and measures, make it difficult to reassemble all the results within the same model. Therefore, it would be good to conduct one large study including male/female young adults and male/female older adults in order to have a better description and understanding of sex specificities.

In relation to our measure of physical activity, although we quantified the physical activity of participants with a validated questionnaire, the proportion of aerobic and resistance activity was not determined in this study. The development of protocols or questionnaires that can address this issue will be beneficial for future investigation on this topic. In addition, objective measures of exercise intensity such as accelerometry, heart rate, or oxygen uptake should be used to complement the questionnaire-based physical activity recording and provide a more comprehensive assessment of exercise intensity. Furthermore, the continuous-wave fNIRS technique has several limitations. First, it has lower spatial resolution and poorer depth penetration than other brain imaging techniques [63]. Secondly, this technique can be influenced by changes in blood flow on extra-cerebral tissue such as extra-cerebral tissue included the scalp, skull, and cerebrospinal fluid [64,65]. In this study, we did not control the superficial skin blood flow but the influence of changes in scalp blood flow is likely to be minimal since participants remained in a seated position during the entire protocol. Thirdly, the fNIRS device used quantifies changes in cerebral oxygenation and not absolute oxygenation. Therefore, group comparisons using this type of fNIRS should be interpreted with caution, as it is possible that in certain protocols, group differences exist already in baseline tissue oxygenation and blood volume, and also optical properties of the brain and superficial layers [63]. In this study, we may have underestimated the impact of physiological noise as we have not performed the filtering of the NIRS signals and detrending of segments. Despite this, our fNIRS results are in accordance with the results of literature. The results obtained in the present study concern only the PFC (not deep brain structures) and not the whole of the brain, since the cognitive task used strongly stimulates this region of interest. Combining the results of NIRS with electroencephalogram (EEG) and functional MRI will provide a more comprehensive understanding of the dynamics of cerebral activation and the other areas of the brain involved. The utilization of a multi-channel NIRS setup can help to observe the oxygenation profiles and activities of other brain regions compared to the two-channel setup in this study.

## 5. Conclusions

In conclusion, this study supports the positive effect of PA on executive performances in healthy young males. The results demonstrated that the active young males performed better (had faster responses) in the executive switching condition than inactive young males. Furthermore, active participants had a larger increase in PFC oxygenation in the switching condition than the inactive participants. Accordingly, fulfilling the physical activity recommendation of 150 MVPA/week may be able to support healthy young males to increase their PFC oxygenation and to perform better in activities that require executive function.

## Figures and Tables

**Figure 1 ijerph-17-01406-f001:**
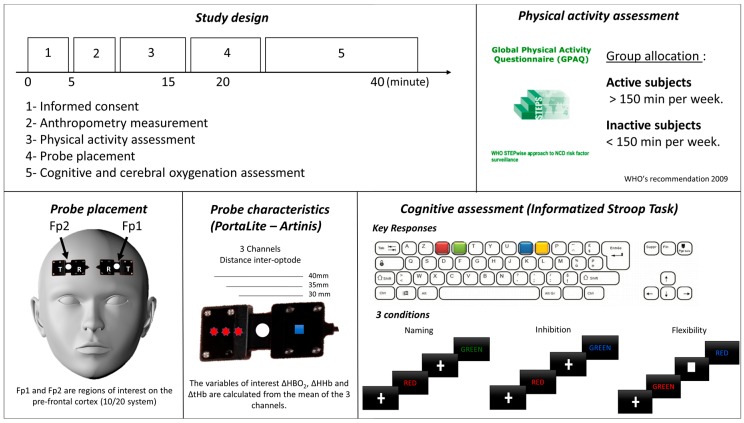
Study design and methodological information.

**Figure 2 ijerph-17-01406-f002:**
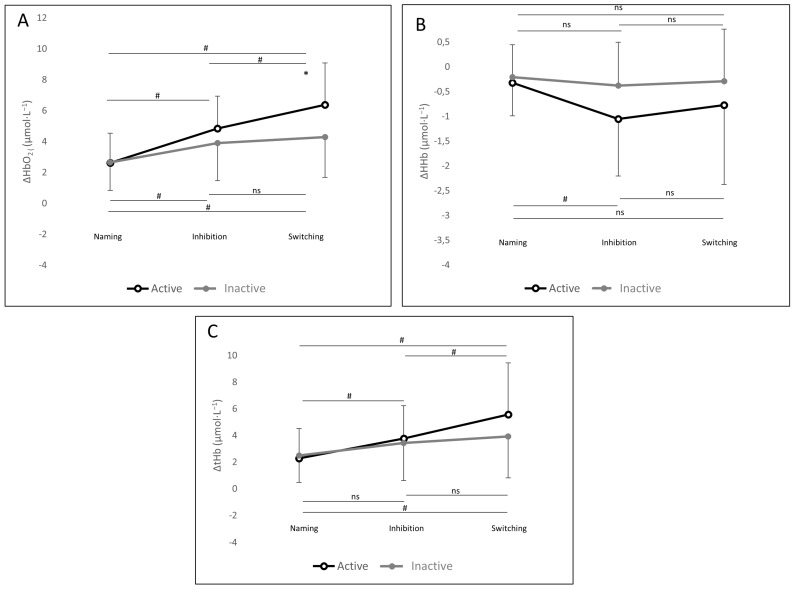
Total cerebral oxygenation (both hemispheres of prefrontal cortex (PFC)) during Computerized Stroop task conditions. (**A**) Change in total oxyhemoglobin (ΔHbO_2_), (**B**) Change in total deoxyhemoglobin (ΔHHb), (**C**) Change in total total hemoglobin (ΔtHb). Data are presented as means and SDs; **p* < 0.05 between active and inactive group. ^#^
*p* < 0.05 between Stroop conditions.

**Figure 3 ijerph-17-01406-f003:**
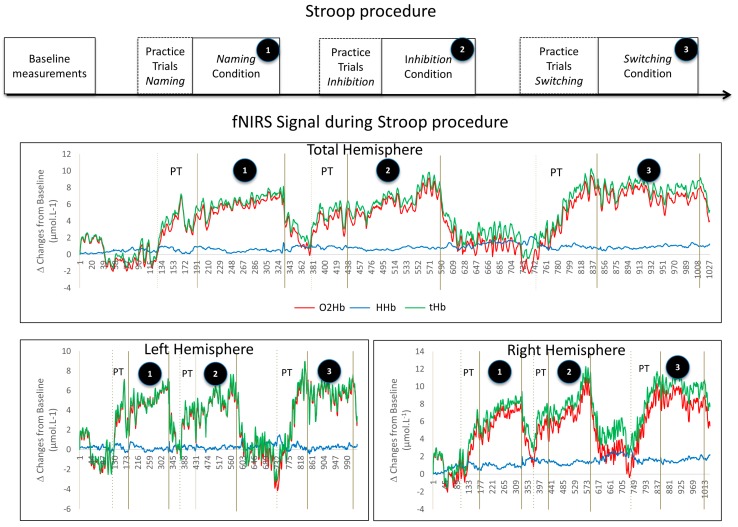
Typical response of cerebral oxygenation during the Stroop procedure. (PT = practice trials).

**Table 1 ijerph-17-01406-t001:** Characteristics of study participants. Data are presented as mean (SD).

	Active (n = 26)	Inactive (n = 30)	Cohen’s *d*
Age (years)	21.8 (2.0)	23.3 (2.5)	0.66
Height (cm)	177.0 (5.2)	180.0 (5.4)	0.56
Weight (kg)	74.4 (15.1)	81.9 (22.2)	0.35
GPAQ (METS·min·week^−1^)	5165.6 (3842.8)	536.0 (275.2) *	1.77
${\dot{V}O}_{2}\mathrm{peak}$ (mL·kg^−1^·min^−1^)	49.8 (8.5)	38.6 (7.2) *	2.90

GPAQ, Global Physical Activity Questionnaire; ${\dot{V}O}_{2}\mathrm{peak}$, peak oxygen uptake. * different from Active (*p* < 0.05).

**Table 2 ijerph-17-01406-t002:** Accuracy and reaction time during Stroop tasks. Data are presented as mean (SD).

	Overall (n = 56)	Active (n = 26)	Inactive (n = 30)	Cohen’s *d* ^+^
**Accuracy (% correct responses)**				
Naming	96.7 (2.2)	96.7 (2.4)	96.7 (2.5)	0.0
Inhibition	97.4 (2.4)	96.5 (2.6)	97.8 (3.1)	0.5
Switching	92.6 (4.2)	91.4 (5.9)	93.7 (4.9)	0.4
**Reaction Time (ms)**				
Naming	605.4 (83.5)	594.4 (105.8)	614.9 (63.2)	0.2
Inhibition	707.2 (91.9)	682.9 (94.0)	728.3 (92.7)	0.5
Switching	958.4 (145.9)	901.3 (137.6)	1007.8 (138.2) *	0.7

* different from active group (*p* < 0.05); ^+^ effect size between active and inactive group.

**Table 3 ijerph-17-01406-t003:** Right and left prefrontal cerebral oxygenation changes from baseline during computerized Stroop tasks. Data are presented as means (SDs).

	Active (n = 26)		Inactive (n = 30)		Cohen’s *d*
■ Naming					
*Right PFC*					
■ ΔHbO_2_ (μmol·L^−1^)	2.29	(2.06)	2.48	(1.99)	0.1
■ ΔHHb (μmol·L^−1^)	−0.44	(0.69)	−0.32	(0.66)	0.2
*Left PFC*					
■ ΔHbO_2_ (μmol·L^−1^)	3.13	(1.90)	2.81	(1.85)	0.2
■ ΔHHb (μmol·L^−1^)	−0.15	(0.75)	−0.14	(0.68)	0.0
■ Inhibition					
*Right PFC*					
■ ΔHbO_2_ (μmol·L^−1^)	4.75	(2.27)	3.54	(2.42)	0.5
■ ΔHHb (μmol·L^−1^)	−1.32	(1.19)	−0.42	(0.83) *	0.9
*Left PFC*					
■ ΔHbO_2_ (μmol·L^−1^)	5.14	(2.22)	4.02	(2.40)	0.5
■ ΔHHb (μmol·L^−1^)	−0.75	(1.25)	−0.38	(0.86)	0.4
■ Switching					
*Right PFC*					
■ ΔHbO_2_ (μmol·L^−1^)	6.16	(2.38)	3.73	(2.67) *	0.8
■ ΔHHb (μmol·L^−1^)	−1.11	(1.70)	−0.26	(1.10)	0.6
*Left PFC*					
■ ΔHbO_2_ (μmol·L^−1^)	6.87	(3.21)	4.45	(2.81) *	0.8
■ ΔHHb (μmol·L^−1^)	−0.37	(1.67)	−0.37	(1.05)	0.0

PFC, Prefrontal cortex; ΔHbO_2_, changes in oxyhemoglobin concentrations; ΔHHb, changes in deoxyhemoglobin concentrations. The results of dual-task condition are the difference from the baseline. * different from active group (*p* < 0.05).

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
