# Peer review of "Cerebral Oxygenation Reserve: The Relationship Between Physical Activity Level and the Cognitive Load During a Stroop Task in Healthy Young Males"

_ijerph, 2020, doi:10.3390/ijerph17041406_

Round 1

Reviewer 1 Report

Page 2, line 88-89: the authors hypothesize that executive performance would be better in participants with a higher physical activity. This is a correlational relationship. Later on, the authors refer to there being a "positive impact of physical activity on performance" -> which is a causal relationship. Their study design and data do not support this causal relationship. I am interested in the authors' opinions on how to either restructure the takeaway from their work, or remove the notion that greater physical activity levels are directly related to executive performance. If they remove the statement and reformat their discussion and data interpretation, it is hard to appreciate the value of the physical activity data and how it relates to performance, besides for interest in designing future studies. I am open to discussion on this topic and look forward to hearing how the authors choose to address this.

Page 3, lines 94-98: it is not evident how the authors selected the number of participants for their study. 

The authors also report that they did not recruit anyone with neurological or cardiovascular disorders. What about cerebrovascular disorders (unless they are lumping cerebrovascular conditions under cardiovascular disease)? These could potentially have a role in oxygenation signals.

RESULTS: are the individual HRFs repeatable within each subject? That is, are the signals produced by each participant for a given task truly representative of their response in PFC?

Page 11, line 386: Without conducting a longitudinal study, it is very difficult to use the collected data herein to support the claim "this study supports the positive effect of PA on executive performances in healthy males".

I commend the authors for carrying out the research and putting together this manuscript. I look forward to hearing their responses to my comments and feedback.

Author Response

Page 2, line 88-89: the authors hypothesize that executive performance would be better in participants with a higher physical activity. This is a correlational relationship. Later on, the authors refer to there being a "positive impact of physical activity on performance" -> which is a causal relationship. Their study design and data do not support this causal relationship. I am interested in the authors' opinions on how to either restructure the takeaway from their work, or remove the notion that greater physical activity levels are directly related to executive performance. If they remove the statement and reformat their discussion and data interpretation, it is hard to appreciate the value of the physical activity data and how it relates to performance, besides for interest in designing future studies. I am open to discussion on this topic and look forward to hearing how the authors choose to address this.

Response: The test used is an ANOVA. An interaction in ANOVA means that a certain outcome (i.e., executive performance or cerebral oxygenation measured by HbO2) is dependent on physical activity level. Such that when comparing people with low physical activity level to people with higher physical activity levels perform better on EF portion of Stroop then non-EF portion of Stroop.

Accordingly, we modified the hypotheses in order to account for your comment.

Page 3, lines 94-98: it is not evident how the authors selected the number of participants for their study. 

Response: Assuming the use of a two way factorial ANOVA, a power analysis (Baussell and LI, 2002) indicated that 28 participants per group produced an > 90% chance of obtaining statistical significance at 0.05 level for the 1.00 ES observed in the study by Dupuy et al. (2015)

The authors also report that they did not recruit anyone with neurological or cardiovascular disorders. What about cerebrovascular disorders (unless they are lumping cerebrovascular conditions under cardiovascular disease)? These could potentially have a role in oxygenation signals.

Response: Yes, we agree that cerebrovascular disorders have a potential to affect cerebral oxygenation signals. We did screen for vascular disorders in participants. We also had participants complete the Montreal Cognitive Assessment (MoCA)  to verify their cognitive status (This was not included  in the article; but all participants passed the threshold of 25/26 points out of 30). It is noteworthy that the MOCA as also been validated in adolescents and young adults. (Pike, N. A., Poulsen, M. K., & Woo, M. A. (2017). Validity of the Montreal Cognitive Assessment screener in adolescents and young adults with and without congenital heart disease. Nursing research, 66(3), 222.)

RESULTS: are the individual HRFs repeatable within each subject? That is, are the signals produced by each participant for a given task truly representative of their response in PFC?

Response: The participants completed the task only one time for each condition. Therefore we did not have data on whether HRFs are repeatable or not.

If we think about whether the PFC oxygenation during Stroop task is affected by the global change of CBF or local change/response in other brain areas. We do not think that was the case as, first, the task was taken in a sitting position (without another task or activity) which normally does not provoke a significant change in CBF. Secondly, we put the fNIRS optodes on areas representing Brodmann’s areas 9 and 10, thus warranting that the signal is representative of PFC activity. Additionally, since we used the Portalite with only 3 channels, this should reduce any crosstalk from other channels that may be closer to other areas in the brain.

Page 11, line 386: Without conducting a longitudinal study, it is very difficult to use the collected data herein to support the claim "this study supports the positive effect of PA on executive performances in healthy males".

Response: Yes, we agree with this statement.

We have changed this statement to : “executive function performance in young healthy males is dependent on physical activity levels”. In this study, participants with higher PA levels performed better on EF tasks than people with lower PA levels.

Reviewer 2 Report

The authors present a study that investigates the effects of physical activity (PA) on executive function in young male adults and they hypothesize that this relationship is mediated by cerebral oxygenation as measured by fNIRS. The premise is interesting and has important implications for encouraging healthy lifestyles and understanding the effect of PA on brain health. However, the methods (i.e., exclusion of female participants), analyses (i.e., unclear statistical analyses), and discussion require a lot of clarification and further reflection.

Major comments

1. Lines 15-16: “However, a large bulk of the knowledge concerning the effects of physical activity level on executive performance in younger adults remains unclear.” – I don’t think this is quite true – there are a lot of studies that indicate a positive relationship between physical activity and cognitive performance (as stated in the previous line). Perhaps the authors are trying to say that they want to examine the mechanisms that underlie this relationship (in which knowledge is lacking)?

2. One of my major concerns is regarding the statistical analyses. The ANOVA that was performed isn’t clear. The abstract states that the authors performed a two-way ANOVA; however, were physical activity and Stroop conditions the independent variables and prefrontal oxygenation as the dependent variable? It also seems that reaction time was included – if so, were multiple ANOVAs conducted? In the Methods section, the authors state, “An analysis of variance (ANOVA) with Bonferroni post-hoc tests were conducted to 200 test the interaction of PA level (active/inactive) by Stroop’s performance (naming, inhibition, 201 switching) and PFC oxygenation.” This makes it seem that it was actually a three-way ANOVA? The Results section refers to multiple ANOVAs, so it is very unclear what statistical analyses were actually performed. Please clarify the statistical analysis and describe it clearly in more detail.

3. Similar to the above, I don’t think the authors performed analyses that directly test their hypothesis. From the description and hypothesis, it appears that the authors think that the mechanisms underlying an improvement in executive function due to PA is increase cerebral oxygenation, but they don’t test this hypothesis directly. This can be tested by performing a mediation analysis and determining whether cerebral oxygenation mediates the relationship between PA and executive function.

4. My other major concern is the exclusion of female participants. The authors note that only male participants were included the study because “PFC functions were influenced by the concentration of estrogen.” This is very problematic. If physical activity differentially affects men and women, then it is even more important to include both groups as the results may not generalize to the population. I understand that sex differences was not a primary concern for this study; however, the entire study (except for the title and some parts in the Discussion) is written as if it generalizes to all young adults, but in fact, it is leaving 50% of those young adults (i.e., young women) out. A reader who does not carefully read the Methods, may incorrectly think that it included men and women as the rest of the study is written to imply that.

In addition, regarding the author’s concern that estrogen may affect PFC functioning, men also have varying levels of estrogen. It is surprising that the authors did not measure or control for this as they say this was the primary reason they did not include women.

It is also important to address this approach in the Discussion. The authors do address this briefly in the limitations, but do not reflect on it other than saying that it will “limit the generalizability of findings in younger adults.” This does not explain why the authors did publish their data in women separately, but did not discuss the influence of estrogen in either study.

This study would be more impactful if it were expanded to include young women, otherwise it does not quite answer main question (i.e., does PA improve executive function through cerebral oxygenation) and add to the literature as the authors claim.

5. It seems like the authors have additional publications that only included female participants – it is unclear why the authors have chosen this approach, rather than combine the results into one study. It is even more surprising as the results appear to be congruous and there is no evidence that estrogen levels differentially affect the results (although the authors did not collect estrogen levels in either of the studies, so that is unclear). This approach of separating data into different studies by sex is surprising and unnecessary.

6. Line 84-85: “Mechanisms should be explored to determine if there are age specificities.” It isn’t clear from this sentence what the authors mean - perhaps the authors are trying to say that there may be age-related effects of PA on executive function?

7. The in-text citations and reference list do not match. There are many in-text citations that do not appear in the references list. Also, some in-text citations have author names, while others are numbered.

Minor comments

1. Lines 14-15: “Many studies have reported that regular physical activity was associated with cognitive performance and more selectively with executive functions.” Related in what way? Positively?

2. Line 47: “improve” should be “improved”

3. Lines 65-66: “the lack of energy for (adenosine triphosphate) ATP synthesis” should be: the lack of energy for adenosine triphosphate (ATP) synthesis.

4. Line 73: The first instance of fNIRS should be defined.

5. Lines 95 and 98: 56 should by Fifty-six and 6 should be six.

6. In addition to the above, there are some grammatical errors that reduce the overall clarity and impact of the paper.

Author Response

1. Lines 15-16: “However, a large bulk of the knowledge concerning the effects of physical activity level on executive performance in younger adults remains unclear.” – I don’t think this is quite true – there are a lot of studies that indicate a positive relationship between physical activity and cognitive performance (as stated in the previous line). Perhaps the authors are trying to say that they want to examine the mechanisms that underlie this relationship (in which knowledge is lacking)?

Response: 

Indeed, in this sentence, we refer to the link between physical activity (PA) and executive function. We consider some studies that have reported no association between PA (and CRF in some studies) and cognitive performance, especially executive function, in young adults. For example the study from Hayes (2016) that found association between CRF and several executive function tests (Trail making, D-KEFS, WAIS-III, and WCST) in older adults (55-82 years) but not in younger adults (18-31 years). We revised this sentence in the introduction to make it clearer.

Before revision :“Many studies have reported that regular physical activity was positively associated with cognitive performance and more selectively with executive functions. However, a large bulk of the knowledge concerning the effects of physical activity level on executive performance in younger adults remains unclear.”

After revision : Many studies have reported that regular physical activity was positively associated with cognitive performance and more selectively with executive functions. However, some studies reported that the association of physical activity on executive performance in younger adults was not as clearly established (as some have demonstrated no relationship) when compared to studies with older adults.”

2. One of my major concerns is regarding the statistical analyses. The ANOVA that was performed isn’t clear. The abstract states that the authors performed a two-way ANOVA; however, were physical activity and Stroop conditions the independent variables and prefrontal oxygenation as the dependent variable? It also seems that reaction time was included – if so, were multiple ANOVAs conducted? In the Methods section, the authors state, “An analysis of variance (ANOVA) with Bonferroni post-hoc tests were conducted to 200 test the interaction of PA level (active/inactive) by Stroop’s performance (naming, inhibition, 201 switching) and PFC oxygenation.” This makes it seem that it was actually a three-way ANOVA? The Results section refers to multiple ANOVAs, so it is very unclear what statistical analyses were actually performed. Please clarify the statistical analysis and describe it clearly in more detail.

Response: The reviewer is correct, for each measure (behavioural: Stroop Reaction time and Accuracy performance) and neural (PFC oxygenation; HbO2, Hbb) a series of 2 (active/inactive) by 3 (naming, inhibition, switching) repeated measures ANOVAs were conducted.

3. Similar to the above, I don’t think the authors performed analyses that directly test their hypothesis. From the description and hypothesis, it appears that the authors think that the mechanisms underlying an improvement in executive function due to PA is increase cerebral oxygenation, but they don’t test this hypothesis directly. This can be tested by performing a mediation analysis and determining whether cerebral oxygenation mediates the relationship between PA and executive function.

Response: We agree with your comment and changed the wording of our initial hypothesis.

4. My other major concern is the exclusion of female participants. The authors note that only male participants were included the study because “PFC functions were influenced by the concentration of estrogen.” This is very problematic. If physical activity differentially affects men and women, then it is even more important to include both groups as the results may not generalize to the population. I understand that sex differences was not a primary concern for this study; however, the entire study (except for the title and some parts in the Discussion) is written as if it generalizes to all young adults, but in fact, it is leaving 50% of those young adults (i.e., young women) out. A reader who does not carefully read the Methods, may incorrectly think that it included men and women as the rest of the study is written to imply that.

Response: As reported in a recent study of Weis (2019), that the brain connectivity in several states are varied across menstrual cycle in females but stable across repeated tests in males, especially when frontal areas are involved. The reason to include only males in this study is to have more control on the effect of estrogen on executive function performance. We did not intend to mislead the reader nor to generalize the effect into all young adults (males and females). Therefore we verified the manuscript and edited sections to clarify this and to not imply that our results generalize to all younger adults.

S. Weis, S. Hodgetts, and M. Hausmann, “Sex differences and menstrual cycle effects in cognitive and sensory resting state networks,” Brain Cogn., vol. 131, pp. 66–73, 2019.

In addition, regarding the author’s concern that estrogen may affect PFC functioning, men also have varying levels of estrogen. It is surprising that the authors did not measure or control for this as they say this was the primary reason they did not include women.

Response: We are agree with you that measuring the estrogen level or concentration will help to determine variation of estrogen in men. However, as discussed in the previous response (4; above), it also reported that brain connectivity in men is stable across repeated tests (in Weis, 2019), and that the effect of estrogen on brain in men is minimal, which is not the case with women.

It is also important to address this approach in the Discussion. The authors do address this briefly in the limitations, but do not reflect on it other than saying that it will “limit the generalizability of findings in younger adults.” This does not explain why the authors did publish their data in women separately, but did not discuss the influence of estrogen in either study.

Response: Although females were not included due to known estrogen effects on cerebral oxygenation (Weis et al., 2019) this is an important group to assess and see how estrogen may influence PA, PFC signal findings in comparison to younger males. The non-inclusion of females in our sample limits our generalizability in younger samples, future work should capture estrogen differences in males and females and see how this influences PA, PFC executive function findings.

Moreover, our previously published data on females were obtained from an independent study, (although some authors are listed in both papers), with specific purposes (the study in females also tested the effect of age and CRF).

This study would be more impactful if it were expanded to include young women, otherwise it does not quite answer main question (i.e., does PA improve executive function through cerebral oxygenation) and add to the literature as the authors claim.

Response: We agree that the specific response of females deserves interest. We already published some data that brings information on this topic. However, information on young males was lacking. This explains why the present study focused specifically on male young adults. Moreover, data collection is closed and is it not possible any more to assess females, although sex comparison would have been of interest. 

5. It seems like the authors have additional publications that only included female participants – it is unclear why the authors have chosen this approach, rather than combine the results into one study. It is even more surprising as the results appear to be congruous and there is no evidence that estrogen levels differentially affect the results (although the authors did not collect estrogen levels in either of the studies, so that is unclear). This approach of separating data into different studies by sex is surprising and unnecessary.

Response: Same comment as above (comment 6 and 7)

It would be good to conduct one large study including YA males/females and OA males/females and truly test the influence of estrogen by sampling estrogen levels in blood tests. Unfortunately, the current study was not designed for this purpose, the goal was to only examine YA males.

6. Line 84-85: “Mechanisms should be explored to determine if there are age specificities.” It isn’t clear from this sentence what the authors mean - perhaps the authors are trying to say that there may be age-related effects of PA on executive function?

Response: Yes, the reviewer is correct. We aimed to raise the point that there may be an age-related effect of PA on executive performance that needs further examination. We have corrected the phrasing to make this clearer.

Before revision : “Mechanisms should be explored to determine if there are age specificities”

After revision :  “The possibility of a specific age-related effects of PA on executive performance should be further explored”

7. The in-text citations and reference list do not match. There are many in-text citations that do not appear in the references list. Also, some in-text citations have author names, while others are numbered.

Response: Thank you for noticing these errors. We have corrected these errors in the manuscript.

Minor comments

1. Lines 14-15: “Many studies have reported that regular physical activity was associated with cognitive performance and more selectively with executive functions.” Related in what way? Positively?

Response: 

We revised this sentence.

Before revision : “Many studies have reported that regular physical activity was associated with cognitive performance and more selectively with executive functions.”

After revision : “Many studies have reported that regular physical activity was positively associated with cognitive performance and more selectively with executive functions.”

2. Line 47: “improve” should be “improved”

Response: Do you mean in this sentence : “Regular PA has also been shown to influence cognitive performance throughout the lifespan, contributing to improvedacademic achievement in children, adolescents or young adults and preventing cognitive decline and dementia in older adults”

3. Lines 65-66: “the lack of energy for (adenosine triphosphate) ATP synthesis” should be: the lack of energy for adenosine triphosphate (ATP) synthesis.

Response: We have revised this sentence accordingly.

4. Line 73: The first instance of fNIRS should be defined.

Response: 

Before revision : “Although the studies in the fNIRS field are growing exponentially, most of the studies that address the relationship between PA and cerebral oxygenation have the goal of measuring these factors in older adults to prevent or reduce cognitive impairment.”

After revision : “Although the studies in the functional near-infrared spectroscopy (fNIRS) field are growing exponentially, most of the studies that address the relationship between PA and cerebral oxygenation have the goal of measuring these factors in older adults to prevent or reduce cognitive impairment.”

5. Lines 95 and 98: 56 should by Fifty-six and 6 should be six.

Response: We revised this as advised.

6. In addition to the above, there are some grammatical errors that reduce the overall clarity and impact of the paper.

Response: We have reviewed the manuscript thoroughly to minimize these types of errors.

Reviewer 3 Report

This manuscript presents cognitive and cerebral oxygenation data from young fit and unfit males. The topic is of great interest to the literature and I think less well defined than even the authors suggest. There is still a relative lack of consensus regarding the contributions of exercise and fitness to cognition, or the mechanisms of a purported relationship. In that sense, this is an important contribution to the literature. 

I have one major concern and a few minor suggestions.

The authors note that "The research effort ... should not be limited to the elderly, but be concerned with all ages of life."  Shouldn't this inclusivity also be extended to sex? The authors state that PFC function is influenced by estrogen? Is there no evidence it's influenced by testosterone, or coffee, or depression, previous nights sleep, time of day? To exclude women from a human research study without a clear sex-based hypothesis in he 21st century is problematic. What makes it even more confusing is that the authors have previously published similar Stroop results in women, and note that the exclusion of women is a limitation. If it was a sex-based hypothesis, it wouldn't be a limitation. If there is not a sex-based hypothesis, I think the results would be much stronger if women were included, even if the analyses were done separately.

Perhaps at the very least, the authors could spend time talking about their and other findings in women to make a more compelling case to look at the sexes separately.

This is a shame because it is a nicely executed study. The data in men are compelling. I really like the added complexity of the Switching task, as I've sometimes struggled with the Stroop in high performing individual. The study is also described clearly and the figures tell a nice story.

A couple other minor comments.  I was always taught to present an interaction effect before main effects because the main effects are difficult to interpret in the presence of an interaction. The authors do the opposite here.  For example I'm not sure the main effects of PA and condition can be readily interpreted since there is an interaction. 

Also, I just want to confirm the author are using a repeated measures ANOVA. That term isn't used but they do reference checking sphericity.

Author Response

The authors note that "The research effort ... should not be limited to the elderly, but be concerned with all ages of life."  Shouldn't this inclusivity also be extended to sex? The authors state that PFC function is influenced by estrogen? Is there no evidence it's influenced by testosterone, or coffee, or depression, previous nights sleep, time of day? To exclude women from a human research study without a clear sex-based hypothesis in he 21st century is problematic. What makes it even more confusing is that the authors have previously published similar Stroop results in women, and note that the exclusion of women is a limitation. If it was a sex-based hypothesis, it wouldn't be a limitation. If there is not a sex-based hypothesis, I think the results would be much stronger if women were included, even if the analyses were done separately.

Response: 

This point has been raised by the second reviewer. Please refer to our responses to comments 4 to 7.

As reported in a recent study of Weis (2019), the brain connectivity in several states are varied across menstrual cycle in women but stable across repeated tests in men, especially when frontal areas are involved. The reason to include only men in this study is to minimize the effect of estrogen on executive function performance.

The limitation that we mentioned is about the generalizability of the results to young adults as we only measured male participants.

We agree with you and reviewer number two that future studies should address both sexes (at a minimum) and have objective measures of estrogen/testosterone levels in order to better assess the influence of these factors. As fNIRS, is a newer emerging area of research, and we were unable to conduct blood samples for this study, we felt that focusing on males could be a first step in understating the neural mechanisms underlying PA, PFC activity and executive function. We will aim to be more inclusive in future samples.

S. Weis, S. Hodgetts, and M. Hausmann, “Sex differences and menstrual cycle effects in cognitive and sensory resting state networks,” Brain Cogn., vol. 131, pp. 66–73, 2019.

Perhaps at the very least, the authors could spend time talking about their and other findings in women to make a more compelling case to look at the sexes separately.

Response: 

We added a specific paragraph in the limitations: “As reported in a recent study by Weis et al. (2019), brain connectivity differs across menstrual cycle in females but remains stable in males, especially when frontal areas are involved. Moreover, information on male young adults is still lacking, while several studies have been published in females. This explains why this study focused on males. However, this approach limits the generalizability of our findings to this specific population. Understanding sex specificities is of great importance. Although several studies have been conducted with males or females separately, differences in participants characteristics (others than sex), as well in tests and measures make it difficult to reassemble all the results within the same model. Therefore, it would be good to conduct one large study including male/female young adults and male/female older adults in order to have a better description and understanding of sex specificities. “

A couple other minor comments.  I was always taught to present an interaction effect before main effects because the main effects are difficult to interpret in the presence of an interaction. The authors do the opposite here.  For example I'm not sure the main effects of PA and condition can be readily interpreted since there is an interaction. 

Response: Our research team is used to reporting all the effects. However, it is true that the interaction effect supersedes the main effects and many authors only report the interaction. We modified several sentences.

Also, I just want to confirm the author are using a repeated measures ANOVA. That term isn't used but they do reference checking sphericity.

Response: Yes this should be clarified in stat analysis

Round 2

Reviewer 1 Report

Revised comments are fine as is. 

Reviewer 2 Report

The authors have addressed my concerns. I have no further comments.

Reviewer 3 Report

Adequately address my concerns. No further comments.